# Evaluation of Wheat Germplasm for Resistance to Leaf Rust (*Puccinia triticina*) and Identification of the Sources of *Lr* Resistance Genes Using Molecular Markers

**DOI:** 10.3390/plants10071484

**Published:** 2021-07-20

**Authors:** Alma Kokhmetova, Shynbolat Rsaliyev, Makpal Atishova, Madina Kumarbayeva, Angelina Malysheva, Zhenis Keishilov, Danna Zhanuzak, Ardak Bolatbekova

**Affiliations:** 1Institute of Plant Biology and Biotechnology, Almaty 050040, Kazakhstan; maki_87@mail.ru (M.A.); madina_kumar90@mail.ru (M.K.); malysheva_angelina@list.ru (A.M.); jeka-sayko@mail.ru (Z.K.); dolphin_969@mail.ru (D.Z.); ardashka1984@mail.ru (A.B.); 2Faculty of Agrobiology, Kazakh National Agrarian Research University, Almaty 050010, Kazakhstan; 3Kazakh Research Institute of Agriculture and Plant Growing, Almalybak 040909, Kazakhstan

**Keywords:** wheat, leaf rust, *Lr* genes, virulence, pathotypes, molecular markers

## Abstract

Leaf rust, caused by *Puccinia triticina* (*Ptr*), is a significant disease of spring wheat spread in Kazakhstan. The development of resistant cultivars importantly requires the effective use of leaf rust resistance genes. This study aims to: (i) determine variation in *Ptr* population using races from the East Kazakhstan, Akmola, and Almaty regions of Kazakhstan; (ii) examine resistance during seedling and adult plant stages; and (iii) identify the sources of *Lr* resistance genes among the spring wheat collection using molecular markers. Analysis of a mixed population of *Ptr* identified 25 distinct pathotypes. Analysis of these pathotypes using 16 Thatcher lines that are near-isogenic for leaf rust resistance genes (*Lr*) showed different virulence patterns, ranging from least virulent “CJF/B” and “JCL/G” to highly virulent “TKT/Q”. Most of the pathotypes were avirulent to *Lr9*, *Lr19*, *Lr24*, and *Lr25* and virulent to *Lr1*, *Lr2a*, *Lr3ka*, *Lr11*, and *Lr30.* The *Ptr* population in Kazakhstan is diverse, as indicated by the range of virulence observed in five different races analyzed in this study. The number of genotypes showed high levels of seedling resistance to each of the five *Ptr* races, thus confirming genotypic diversity. Two genotypes, Stepnaya 62 and Omskaya 37, were highly resistant to almost all five tested *Ptr* pathotypes. Stepnaya 62, Omskaya 37, Avangard, Kazakhstanskaya rannespelaya, and Kazakhstanskaya 25 were identified as the most stable genotypes for seedling resistance. However, most of the varieties from Kazakhstan were susceptible in the seedling stage. Molecular screening of these genotypes showed contrasting differences in the genes frequencies. Among the 30 entries, 22 carried leaf rust resistance gene *Lr1*, and two had *Lr9* and *Lr68. Lr10* and *Lr28* were found in three and four cultivars, respectively. *Lr19* was detected in Omskaya 37. Two single cultivars separately carried *Lr26* and *Lr34*, while *Lr37* was not detected in any genotypes within this study. Field evaluation demonstrated that the most frequent *Lr1* gene is ineffective. Kazakhstanskaya 19 and Omskaya 37 had the highest number of resistance genes: three and four *Lr* genes, respectively. Two gene combinations (*Lr1*, *Lr68*) were detected in Erythrospermum 35 and Astana. The result obtained may assist breeders in incorporating effective *Lr* genes into new cultivars and developing cultivars resistant to leaf rust.

## 1. Introduction

Central Asia, including Kazakhstan, is a significant player in regional and global food security, producing most of the grain traded in the region, with total area sown to wheat in Kazakhstan representing over 85% of total cereal production [1]. One of the main reasons for the reduction in the yield of wheat in Kazakhstan is the disease with airborne infection. Dominant position, as a part of the pathogenic complex of wheat in Kazakhstan, is taken by rusts (yellow, stem, and leaf rust) [2,3,4,5], as well as leaf spot diseases (tan spot and Septoria) [6,7,8,9,10,11].

Wheat leaf rust fungus, *Puccinia triticina* (*Ptr*), is found in major wheat-growing regions of the world and is a leading cause of yield loss in wheat. Populations of *P. triticina* are highly variable for virulence to resistance genes in wheat and adapt quickly to resistance genes in wheat cultivars [12]. It caused serious damage to both yield and quality. On average, the disease causes 21.5% of yield losses in wheat globally [13].

Wheat-growing regions of Kazakhstan have been facing frequent occurrences of leaf rust epidemics. Between 2001 and 2009, North Kazakhstan suffered leaf rust epidemics that occurred five times (2002, 2003, 2005, 2007, and 2009), and the yield loss due to leaf rust has been reported to range from 10% to 50% in the most susceptible cultivars [3,14]. Leaf rust in Kazakhstan is spread from 4% to 61% of the surveyed area, which represents approximately from 0.5 to 3 million hectares. Review on leaf rust incidence, virulence, and breeding in northern Kazakhstan and Siberia [15] found that the pathogen affects up to 5 Mha of spring wheat on average one year out of four with yield losses of 25% to 30%. The use of genetically rust-resistant cultivars is considered to be the most efficient, cost-effective, and environmentally safe method for disease control.

Seedling resistance (all-stage resistance) and adult plant resistance (APR) are the main categories used to describe the reaction of wheat to rust [16]. APR genes provide resistance to individual or all pathotypes of the fungus [17]. Race-nonspecific APR-resistance provides partial (or slow-rusting) resistance [18]. Currently, 79 *Lr* genes for resistance to leaf rust have been identified [19]. Some APR genes are race-specific (*Lr12*) [20], others are race non-specific (*Lr34*, *Lr46*, *Lr67*, *Lr68*, *Yr36*, and *Sr2*) [21]. Race-specific APR genes *Lr34* [22], *Lr67* [23], and *Yr36* [24] were cloned; it was found that they encode the ABC and hexose transporters and the kinase START gene, respectively. New mechanisms of resistance of these classes of genes lead to pleiotropism and long-term resistance. Several race-specific *Lr* genes belong to the NBS-LRR (nucleotide-binding site leucine-rich repeat) class and encode receptor proteins on the signal transduction pathway that appears in response to pathogen exposure [25].

Diversity in *Lr* genes in commercial cultivars could play an important role in managing frequent leaf rust epidemics in the region. Previous studies carried out in Kazakhstan showed that emergence of new virulent races of the pathogen leads to the ineffectiveness of a number of *Lr* genes. Genes *Lr9*, *Lr10*, *Lr19*, *Lr34*, *Lr37*, and *Lr68* are still effective, while *Lr1* has lost its effectiveness [26,27]. The comparative study of population structure in the West Asian region of Russia and northern Kazakhstan revealed high genetic similarity in virulence and phenotypic composition between Omsk and North Kazakhstan, Omsk and Chelyabinsk populations [28]. The study on leaf rust incidence, virulence, and breeding in northern Kazakhstan and Siberia showed the absence of virulence for genes *Lr9* and *Lr24* in the leaf rust population and a low proportion of isolates with virulence to *Lr11*, *Lr16*, *Lr18*, and *Lr28*. The field observations indicated that genes *Lr28* and *Lr36* provide resistance [15]. Avirulence to *Lr19* and *Lr24* and virulence to *Lr3a*, *Lr3bg*, *Lr3ka*, *Lr14a*, *Lr14b*, *Lr16*, *Lr17*, and *Lr30* was shown in a more detailed study of 2016 targeting these regions. The proportion of isolates virulent to *Lr1*, *Lr2a*, *Lr2b*, *Lr2c*, *Lr9*, *Lr15*, *Lr18*, *Lr20*, and *Lr26* varied depending on location [29].

Some of the *Lr* genes are closely linked to other resistance genes, e.g., *Lr19/Sr25; Lr26/Yr9/Sr31/Pm8*, *Lr37/Yr17/Sr38*, and *Lr34/Yr18/Pm38*, that are still effective or represent great interest as donors of valuable agronomic traits in Kazakhstan [4].

The leaf rust population showed an absence of virulence for genes *Lr9* and *Lr24* and a low proportion of isolates with virulence to *Lr11*, *Lr16*, *Lr18*, and *Lr28*. The field observations indicated that genes *Lr28* and *Lr36* provide resistance.

Previous studies have reported variations in *Ptr* populations in Kazakhstan [4,27,30]. It is therefore necessary to periodically evaluate resistant cultivars and advanced breeding lines against *Ptr* races in order to monitor resistance breakdown and plan the replacement of susceptible cultivars with resistant cultivars.

Identification and selection of resistant genes through gene postulation and other plant protection and breeding strategies are time-consuming and cannot be employed if no different fungal isolates are available [31,32]. The molecular marker technology—the most accurate and efficient tool to screen wheat material against various genes, conferring resistance to rust pathogens, and for developing disease-resistant cultivars—is needed to overcome these problems [31,32,33,34].

Understanding the nature of resistance and prevention of genetic erosion leading to a rapid efficiency loss of used genes requires screening of new material. The screening should include the identification of resistance genes using molecular markers, as well as the study of the response of seedlings and adult plants of commercial varieties and advanced breeding lines to the *Ptr* pathotypes.

There is an understanding of adult plant resistance to leaf rust in winter wheat [4,27], but there is limited information available on *Lr* genes present in commercial cultivars and advanced breeding lines of spring wheat from Kazakhstan.

Several recent reports listed a number of released cultivars and advanced breeding lines of wheat in Kazakhstan and Russia that were resistant to leaf rust [27,30,35], but their reactions to the diverse *Ptr* races of Kazakhstan is not known. This study aims to: (i) determine variation in *Ptr* population using races from East Kazakhstan, Akmola, and Almaty regions of Kazakhstan; (ii) examine seedling and adult plant stage resistance; and (iii) identify the sources of resistance among the spring wheat collection using molecular markers.

## 2. Results

### 2.1. Races of Puccinia triticina and Their Virulence Pattern

Use of 16 Tc*Lr*-lines helped to identify 25 virulence phenotypes of *P*. *triticina* from leaf rust-infected leaf samples that were collected in East Kazakhstan (East), Akmola (North), and Almaty (Southeast) regions of Kazakhstan during the 2019 and 2020 seasons (Table 1). The northern population of leaf rust is more numerous (12 pathotypes) and more virulent in comparison to the eastern (8 pathotypes) and the southeastern (4 pathotypes) population. The virulence patterns of the pathotypes ranged from least virulent “CJF/B” and “JCL/G” (virulent on 5 of 16 differentials) to highly virulent “TKT/Q” (virulent on 13 of the 16 differentials). The isogenic lines with *Lr24* and *Lr25* were immune to 23 and 24 pathotypes, respectively. Tc*Lr*-lines with *Lr9* and *Lr19* were immune to 19 of 25 pathotypes.

The average value of the virulence of pathotypes from the Akmola region was 61.0%, which is 2.7% more than the virulence of the eastern population (58.3%) and 25% more than the southeastern population of *P. triticina* (36%).

There were some pathotypes with high virulence to *Lr*-lines in the eastern population of *P. triticina*. Thus, the virulence of the TKT/Q and TRT/G pathotypes were 81.3% and 75.0%, respectively. The southeastern leaf rust population contains four pathotypes with virulence from 31.3% to 43.7%.

Almost all test clones were avirulent to *Lr9*, *Lr19*, *Lr24*, and *Lr25*. Thatcher isogenic lines with these genes show high efficiency for many of the studied leaf rust pathotypes. Accordingly, the sources of these genes protect spring wheat from many leaf rust pathotypes. Most of the pathotypes were virulent to Tc-lines with *Lr1*, *Lr2a*, *Lr3ka*, *Lr11*, and *Lr30.* Isolates differed in their virulence to *Lr3a*, *Lr3bg*, *Lr10*, *Lr16*, *Lr17*, *Lr18*, *Lr20*, *Lr26*, and *Lr29* (Appendix A).

Some isogenic lines (*Lr10*, *Lr23*, *Lr26*, *Lr29*) in most cases exhibit intermediate resistant reactions (2, 2+), which quickly turn into susceptibility during plant development and/or at the slightest change in plant growth conditions. Consequently, the sources of these genes are inappropriate to use in breeding for immunity due to the variability of their resistance response.

### 2.2. Seedling Test

Spring wheat germplasm was evaluated for response to *P. triticina* pathotypes. A complete list of this plant material, its origin, and leaf rust reaction are given in Table 2. The seedling reactions of 30 wheat genotypes to the 5 pathotypes of *Ptr* differed greatly. The wheat genotypes showed arrays of patterns in their responses to the five pathotypes. The most commercial spring wheat varieties in Kazakhstan were generally susceptible to leaf rust pathotypes. The varieties Stepnaya 62 and Omskaya 37 were resistant to almost all tested pathotypes of leaf rust. Omskaya 36 was resistant to two of the five pathotypes. Avangard, Kazakhstanskaya rannespelaya, and Kazakhstanskaya 25 were resistant to one pathotype of leaf rust.

The five pathotypes showed arrays of virulence patterns across 30 wheat genotypes (Table 2). Based on infection type across 30 genotypes, SBR/H was the most virulent, followed by SBP/C and KHT/B. The pathotypes THT/B and QBQ/G were the least virulent. The analysis based on reactions of all five pathotypes showed that Stepnaya 62 and Omskaya 37 were the most stable resistant genotypes.

### 2.3. Field Evaluation

Arrays of variation for resistance to leaf rust under field conditions (Table 2) were present. Most wheat varieties (73.33%) were susceptible to leaf rust in the field. Among 30 genotypes, 7 (Akmola 2, Astana 2, Kazakhstanskaya 25, Stepnaya 62, Omskaya 37, Tertsiya, and Chelyaba jubilejnaja) were considered as resistant (≤20% disease severity) in the adult plant stage under field conditions in Kazakhstan (Table 2). Four genotypes showed ≤5% disease severity in both years. Among the genotypes with the most stable resistance in the seedling stage, Stepnaya 62 and Omskaya 37 were resistant in the adult plant stage under field conditions in both years.

### 2.4. Identification of Leaf Rust Resistance Genes Using Molecular Markers

The occurrence of known *Lr* genes in the 30 wheat cultivars is shown in Table 2. Molecular screening of these genotypes showed contrasting differences in the frequencies of these genes. The most frequent *Lr* gene, identified in the material studied individually or in combination, was *Lr1.* The expected marker fragment associated with *Lr1* was found in 22 of the 30 cultivars, including Akmola 2, Almaken, Astana, Astana 2, Bayterek, Zhenis, Kazakhstanskaya rannespelaya, Kazakhstanskaya 19, Kazakhstanskaya 25, Karagandunskaya 70, Lyazzat, Pavlodarskaya 93, Stepnaya 2, Tselina 50, Tselinnaya 3C, Erythrospermum 35, Erythrospermum 841, Stepnaya 62, Omskaya 37, Tertsiya, Chelyaba jubilejnaja, and Saratovskaya 29. The marker for *Lr9* was found in two Russian cultivars, Tertsiya and Chelyaba jubilejnaja, in this study. The marker linked to *Lr10* was found in three Russian cultivars, Albidum 28, Omskaya 37, and Saratovskaya 29. The 1BL.RS translocation carrying *Lr26* and *Lr19* were present in one Russian cultivar, Omskaya 37. Two single cultivars separately carried *Lr26* and *Lr34*, while *Lr37* was not detected in any genotypes in this study. The marker linked to *Lr28* was found in four stocks (Bayterek, Kazakhstanskaya 19, Saratovskaya 29, and Shortandinskaya uluchshennaja 95). The marker linked to *Lr34* was found only in h cv. Kazakhstanskaya 19. Cultivars with the genes *Lr37* and *Lr19* were not identified. The marker closely linked to *Lr68* was found only in two cvs. Astana and Karagandinskaya 70.

Two cultivars had the highest number of resistance genes: Omskaya 37 (*Lr1*, *Lr10*, *Lr19*, and *Lr26)* and Kazakhstanskaya 19 (*Lr1*, *Lr28*, *Lr34)*, although *Lr1* was not effective against the most leaf rust pathotypes studied in this research. Two gene combinations were detected in Kazakh cvs. Erythrospermum 35 (*Lr1*, *Lr68*) and Astana (*Lr1*, *Lr68).* From all investigated cultivars, six Kazakh wheats (Avangard, Kazakhstanskaya 10, Karabalykskaya 92, Nargiz, Omskaya 36, and Ulbinka 25) and susceptible cultivar Morocco failed to show evidence of any of the nine *Lr* markers tested.

## 3. Discussion

Leaf rust is a perennial problem for spring and winter wheat in Central Asia, including Kazakhstan [14,15,35], and *Ptr* populations are diverse and highly virulent [27,30].

Previous studies have reported variations in *Ptr* populations in Kazakhstan [4,27,30]. It is therefore necessary to periodically evaluate resistant cultivars and advanced breeding lines against *Ptr* races in order to monitor resistance breakdown and plan the replacement of susceptible cultivars with resistant ones.

This study provides additional information by presenting 25 diverse *Ptr* races with wide virulence patterns on 16 international differentials of leaf rust and 30 genetically diverse genotypes in both seedling and adult plant stages. Each of the 25 races showed diverse reaction patterns on the wheat genotypes, with virulence varying from resistant to highly susceptible. *Lr25* and *Lr24* were the most effective genes in Kazakhstan. In this study, the isogenic lines with *Lr24* and *Lr25* were immune to 23 and 24 pathotypes, respectively. Tc*Lr*-lines with *Lr9* and *Lr19* were immune to 19 of 25 pathotypes from Kazakhstan. Most of the pathotypes were avirulent to *Lr9*, *Lr19*, *Lr24*, and *Lr25* and virulent to *Lr1*, *Lr2a*, *Lr3ka*, *Lr11*, and *Lr30*. The study of the virulence of the Russian South Ural population of *P. triticina* showed that all isolates were avirulent to Tc-lines with gene *Lr16*, *Lr19*, *Lr24*, *Lr28*, and *Lr29* and virulent to *Lr1*, *Lr3a*, *Lr3bg*, *Lr3ka*, *Lr14a*, *Lr14b*, *Lr17*, and *Lr18* [30]. This suggests that at least two genes (*Lr19* and *Lr24*) have been found to be effective against pathogen populations from both Kazakhstan and Russia.

The number of genotypes showed high levels of seedling resistance to each of the 5 *Ptr* races, thus confirming genotypic diversity. Two genotypes (Stepnaya 62 and Omskaya 37) were highly resistant to almost all five tested *Ptr* pathotypes, while Omskaya 36 was resistant to more than two races. Therefore, Stepnaya 62, Omskaya 37, Avangard, Kazakhstanskaya rannespelaya, and Kazakhstanskaya 25 were identified as the most stable genotypes for seedling resistance. Omskaya 37 also showed a stable type of reaction at the seedling stage to Russian isolates [28]. However, most of the varieties from Kazakhstan were susceptible in the seedling stage.

In previous studies, the sources of *Lr* resistance genes (*Lr19*, *Lr26*, *Lr37*, *Lr34*, *Lr1*, *Lr26*, *Lr34*, *Lr10*, *Lr37*, *Lr19*, and *Lr68* genes) were identified in winter wheat breeding material [4,36,37,38]. In this research, molecular screening of spring wheat cultivars showed contrasting differences in the frequencies of nine important *Lr* genes. Among the 30 entries, 22 carried leaf rust resistance gene *Lr1*, 6 had *Lr19*, 2 had *Lr9*, and *Lr68*, *Lr10*, and *Lr28* were found in 3 and 4 cultivars, respectively. Two single cultivars separately carried *Lr26* and *Lr34*, while *Lr37* was not detected in any genotypes in this study. Field evaluation demonstrated that the most frequent *Lr1* gene to be ineffective. Kazakhstanskaya 19 and Omskaya 37 had the highest number of resistance genes, three and four *Lr* genes, respectively. Two gene combinations (*Lr1* and *Lr68*) were detected in Erythrospermum 35 and Astana.

The wheat genotypes from Kazakhstan and Russia differed greatly in leaf rust severity recorded at the adult plant stage in the field in Kazakhstan. This supports previous reports on varietal resistance and variation among *Ptr* populations in Kazakhstan [27] and in Russia [30]. However, a high genetic similarity was shown in virulence and phenotypic composition between the Omsk (Russia) and North Kazakhstan [28]. This indicates the possibility of joint breeding programs to improve leaf rust resistance in these countries. Several cultivars (Stepnaya 62, Omskaya 37, Avangard, Kazakhstanskaya rannespelaya, and Kazakhstanskaya 25) showed low leaf rust severity, suggesting their potential value as sources of resistance. The resistance of Omskaya 37 was due to the rye translocation 1AL.1RS in combination with ineffective *Lr1* and *Lr10* and, possibly, by the presence of unknown genes. Kazakhstanskaya 19 showed field resistance provided by partial resistance gene *Lr34* in combination with *Lr1*, *Lr28*. These findings support previous reports that leaf rust resistance improved wheat germplasm are becoming increasingly available in Kazakhstan [4,14].

A number of varieties (Akmola 2, Chelyaba jubileynaya, Astana 2, and Kazakhstanskaya 19) were susceptible in the seedling stage but moderately resistant in the adult plant stage. Such varieties are valuable in terms of potential sources of adult plant resistance [39]. If the presence of both seedling stage resistance and adult plant resistance (APR) and only APR represent major and minor gene control of resistance, respectively, the set of 30 genotypes of this study are indicative of both types of resistance.

The *Ptr* population in Kazakhstan is diverse, as indicated by the range of virulence shown by the five different races analyzed in this study. Wheat cultivars possess a range of variability for response to *Ptr* races, and a number of genotypes differed in their level of disease severity in Kazakhstan, suggesting that *Ptr* populations differ in various regions of Kazakhstan. This study identified some wheat genotypes highly resistant to leaf rust that may contribute to the improvement of leaf rust resistance. The cultivation of new leaf rust-resistant varieties could help reduce disease epidemics in Kazakhstan. Resistant genotypes could also be used as improved parents in crossing programs to develop new varieties.

## 4. Materials and Methods

### 4.1. Plant Material

The object of the study was represented by a collection of 30 spring wheat *Triticum aestivum* entries, including 22 registered cultivars from Kazakhstan and 8 cultivars from Russia, which were evaluated for *Puccinia triticina* resistance in greenhouse studies and field experiments. This germplasm is produced or used in breeding programs of Kazakhstan. The highly susceptible control cultivar Morocco as well as the near-isogenic lines (NILs) of cv. Thatcher was also used in both seedling and field tests.

### 4.2. Experimental Site

Evaluation of field resistance to leaf rust was carried out under conditions of the Kazakh Research Institute of Agriculture and Crop Production (KazNIIZiR), (Almalybak, 43°13’09″ N, 76°36’17″ E, Almaty region) in Southeast Kazakhstan, Almaty region, during 2019 and 2020 cropping seasons. Experiments were conducted with a completely randomized design with two replicates in 1 m^2^. The leaf rust susceptible cultivar Morocco was planted in every 10th row and as a spreader border around the nursery to ensure uniform infection. Fertilizer treatments, 60 and 30 kg/ha of N and P_2_O_5_, respectively, and other management practices corresponded to those normally recommended for the region [40]. Annual rainfall ranged from 332 to 644 mm during the two years. Experimental plants were sown in 1 m^2^ plots in mid-April every two experimental years. Weather conditions in Almaty in 2019 and in 2020 were favorable for the development of leaf rust, and the infection on susceptible checks reached 20S and 40S, respectively; however, there was a severe late development of leaf rust reaching 80% on susceptible check Morocco. The growing seasons were favorable for pathogen infection and disease development. Mean daily temperature and relative humidity showed similar trends in both years. The average maximum air temperature for mid-May in 2019 and 2020 reached 31.3 and 32.5 °C, respectively. From April to June 2019, the mean daily temperature was 11.4, 16.6, and 21.6 °C, respectively, and in 2020, 11.4, 16.6, and 21.8 °C. From April to June 2019, the monthly rainfalls and average relative humidity (RH) were 168, 39, and 72 mm, and 59.5%, respectively, and in 2020, 140, 74, and 30 mm, and 57.3% (www.pogodaiklimat.ru/monitor.php accessed 15 June 2021), conditions highly conducive for leaf rust infection and development.

### 4.3. Race Identification

Race identification was performed under controlled greenhouse conditions at the Research Institute for Biological Safety Problems (RIBSP), Gvardeysky, Zhambyl region, Kazakhstan.

Leaf samples infected with leaf rust *Puccinia triticina* were randomly collected during the main spring wheat-growing season in East Kazakhstan, Akmola, and Almaty regions of Kazakhstan. Eighty to 100 rust-infected leaves with sporulating pustules were collected from the research stations and fields of farmers from susceptible cultivars (Morocco, Pmayat Azieva, etc.). The diseased leaf samples were pressed in the folds of newspaper, placed in an envelope, and stored at 4 °C until further analysis.

The virulence codes for the isolates were based on the three-letter nomenclature of Long and Kolmer (1989) [41], with the addition of four sets of four differential lines, for a total of four letters that describe virulence to 16 differentials. Virulence phenotypes determined on the set of 16 differential lines were binary encoded with 0 and 1 for avirulence and virulence, respectively. Set 1: *Lr1* (RL6003), *Lr2a* (RL6000), *Lr2c* (RL6047), and *Lr3* (RL6002); set 2: *Lr9* (RL6010), *Lr16* (RL6005), *Lr24* (RL 6064), and *Lr26* (6078); set 3: *Lr3ka* (RL6007), *Lr11* (RL6053), *Lr17* (RL6008), and *Lr30* (RL6049); set 4: *Lr10*
*(*RL6004), *Lr18*
*(*RL6009), *Lr21*
*(*RL6043), *Lr23*
*(*RL6012*)*. Cultivar Thatcher was included as a susceptible control. Reaction types of 16 differentials were encoded and designated by a letter using the hexadecimal code according to the corresponding binary quadruple. Then each isolate was given a four-letter code (one letter for each set of four differentials), as adapted from the North American nomenclature for virulence in *Puccinia triticina* [41]. Infection types (IT) of the twenty-five isolates to 16 Thatcher lines that are near-isogenic for leaf rust resistance genes are given in Table 1 and avirulence/virulence profiles.

### 4.4. Multiplication and Preservation of Inoculum

The inoculum of *Puccinia triticina* was multiplied and maintained on the susceptible cultivar Morocco. Seeds of Morocco were sown in 11 × 11 × 15 cm plastic pots and placed at room temperature until germination. Upon germination, seedlings were dislocated to the glasshouse under temperatures of 25–30 °C and 19–21 °C during day and night, respectively. The two-leaf stage seedlings of Morocco were disinfected with Maleic hydrazide (5 mg in 50 mL of water per pot) [42] and inoculated with spores from the infected leaf samples. Four pots of Morocco (five plants per pot) were inoculated with urediniospores from individual rust samples. Inoculated seedlings were incubated in a dew chamber for 24 h at 18–20 °C and 90% humidity before being dislocated to the glasshouse at temperatures of 18–24 °C (day) and 19–21 °C (night). Pustules of leaf rust appeared on the leaves 8–10 days after inoculation, from which inoculum was collected on the 14th day using a mechanical cyclone collector in a zero-size capsule. The inoculum was then preserved in a vacuum glass vial and later transferred to a refrigerator (+4 °C) until further use. A separate collector was used for each isolate and multiplication of culture. Spore collection, storage, and reproduction were then conducted in accordance with the methods of Roelfs et al. [43]. Spores of *P. triticina* were used to determine the pathotypes of leaf rust isolated from wheat leaves in the different regions of Kazakhstan.

### 4.5. Single Spore Culture

Morocco seedlings (8–9 days old) were inoculated by spraying the urediniospores previously increased and suspended in light paraffin mineral oil (70 ether: 30 oil). Plants were dried for 1 h before they were placed in a dew chamber overnight at 18–20 °C and then transferred to the greenhouse, where temperatures were maintained at 18–24 °C and 19–21 °C during the day and night, respectively. Seven days after inoculation, leaves were trimmed with scissors so that just a single uredium remained on the trimmed upper edge of the leaves and preserved with purity [43].

### 4.6. Virulence Analysis

To study the resistance of wheat germplasm, the pathotypes representing different regions (1 pathotype from the east, 2 from the north, 2 from the southeast) and possessing approximately average virulence to isogenic lines were used. Seedlings were grown in a greenhouse at 18–20 °C, with 16 h of supplementary lighting. The seedlings were inoculated with individual *P. triticina* isolates with virulence phenotypes QBQ/G, SBR/H, KHT/B, SBP/C, and THT/B 7 d after planting when the primary leaves were fully emerged. Urediniospores of each of 25 isolates (5 × 10^3^ spores) were spray inoculated onto a differential host series consisting of 16 wheat single-gene near-isogenic lines known to possess resistance genes (*Lr*) in a Thatcher genetic background [41]. The inoculated seedlings were air-dried for at least 30 min and were then placed overnight in a mist chamber at 18 °C and 100% RH. After a period of 12 h of high humidity, inoculated differential lines were placed in a temperature-controlled climatic chamber (20 ± 2 °C, 16 h light/8 h dark). IT was scored on the 0 to 4 scale 10–12 d after inoculation [44]. IT 0 had no hypersensitive flecks, necrosis, or uredinia, IT “;” had distinct hypersensitive flecks; IT 1 had small uredinia surrounded by necrosis; IT 2 had small uredinia surrounded by distinct chlorosis; IT 3 had moderate size uredinia without distinct chlorosis, and IT 4 had large uredinia without distinct chlorosis, with larger and smaller uredinia for each IT were indicated by appending + or −, respectively. Isolates with infection types 0–2 and 3–4 were assumed to be avirulent and virulent, respectively. Leaf rust resistance gene postulations were determined based on the similarity of the Thatcher line IT to the entry, as described in Oelke and Kolmer (2004) [45].

The wheat germplasm was evaluated for *Lr* genes using similar methods as described above. Plant reaction to leaf rust at the seedling stage was evaluated to the same five *P. triticina* isolates QBQ/G, SBR/H, KHT/B, SBP/C, and THT/B.

### 4.7. Field Evaluation of Adult Plant Resistance

Field plots were inoculated with mixed races of *Ptr* obtained from 80 to 100 random infected leaf samples collected from the main spring wheat-growing areas of Kazakhstan. Sampling of spores, their storage, and reproduction was carried out according to the methods of Roelfs et al. [43]. The inoculum was multiplied in the greenhouse on cultivar Morocco, and the collected urediniospores were inoculated by a spore: talc mixture (1:100, 20 mg/m^2^) applied in the tillering stage in spring.

Infection type and severity data were recorded on flag leaves in late May and early June when plots were at boot and milk stages, respectively. The time of second evaluation was also determined when rust severity on the susceptible control Morocco reached 60–80%. Scoring of leaf rust symptoms was performed according to the method developed at the CIMMYT [43]. The five infection types (IT) were: 0—immune; R—resistant; MR—moderately resistant; MS—moderately susceptible; and S—susceptible. Partial resistance in the field was evaluated at boot and milk stages, respectively, using the modified Cobb scale [46]. Leaf rust severities were recorded using three replications, and the means of the replicated data were calculated.

### 4.8. DNA Extraction and Detection of Lr Genes with Molecular Markers

Genomic DNA was extracted from fresh leaves of single plants at the two-leaf seedling stage for each genotype using the CTAB method [47]. The presence of molecular markers to resistance genes *Lr1* (pTAG), *Lr9* (J13), *Lr10* (Fi.2245/Lr10-6/r2), *Lr19/Sr25* (PSY1-EF), *Lr26/Sr31/Yr9/Pm* (SCM9), *Lr28* (Wmc313), *Lr34/Sr57/Yr18* (csLV34), *Lr37/Sr38/Yr17* (Ventriup/LN2), and *Lr68* (csGS) was determined as described by Feuillet et al. (1995) [48], Schachermayr et al. (1994) [49], Schachermayr et al. (1997) [50], Zhang and Dubcovsky (2008) [51], Weng et al. (2007) [52], Vikal et al. (2004) [53], Lagudah et al. (2006) [54], Helguera et al. (2003) [55] and Herrera-Foessel et al. (2012) [56] (Appendix A). Primers and annealing temperature conditions of polymerase chain reaction (PCR) were carried out as described for each *Lr* gene in the references (Appendix A). PCR reactions were performed in a Bio-Rad T100TM Thermal Cycler (Bio-RAD, Hercules, CA, USA). The PCR mixture (25 µL) contained 2.5 µL of genomic DNA (30 ng), 1 µL of each primer (1 pM/µL) (Sigma Aldrich, St. Louis, MO, USA), 2.5 µL of dNTP mixture (2.5 mM, dCTP, dGTP, dTTP and dATP aqueous solution) (ZAO Sileks, Russia), 2.5 µL MgCl_2_ (25 mM), 0.2 µL Taq polymerase (5 units µL) (ZAO Sileks, Russia), 2.5 µL 10X PCR buffer and 12.8 µL ddH20. PCR amplification was performed with a Mastercycler (Eppendorf, Hamburg, Germany) with initial denaturation at 94 °C for 3 min, 45 cycles: 94 °C for 1 min, annealing at 60 °C for 1 min, 72 °C for 2 min, and final elongation at 72 °C for 10 min. The amplification products were separated on 2% agarose gel in TBE buffer (45 mM Tris-borate, 1 mM EDTA, pH 8) [57] with the addition of ethidium bromide. To determine the length of the amplification fragment, a 100-bp DNA ladder (Fermentas, Vilnius, Lithuania) was included. Results were visualized using the Gel Documentation System (Gel Doc XR+, BIO-RAD, Hercules, CA, USA).

## Figures and Tables

**Table 1 plants-10-01484-t001:** Virulence of the 25 pathotypes of *P.*
*triticina* from different regions of Kazakhstan, determined using a subset of near-isogenic differentials in a Thatcher background.

Pathotype	Virulence Formula (Avirulent/Virulent)	Response of *Lr* Genes (%)		Frequencies of Pathotypes
		R	S	
	East Kazakhstan region (East)			
KGQ/B	*Lr* 1,9,24,26,17,30,19,20,25,29/*Lr*2a,2c,3,16,3ka,11	62.5	37.5	8.5
TKT/Q	*Lr*9,25,29/*Lr*1,2a,2c,3,16,24,26,3ka,11,17,30,19,20	18.7	81.3	9.5
DCN/H	*Lr*1,2a,3a,9,16,24,11,30,19,25/*Lr*2c,26,3ka,17,20,29	62.5	37.5	8.0
RCP/G	*Lr*2c,9,16,24,11,19,25,29/*Lr*1,2a,3a,26,3ka,17,30,20	50.0	50.0	24.5
SQT/Q	*Lr*3,24,26,25,29/*Lr*1,2a,2c,9,16,3ka,11,17,30,19,20	31.3	68.7	14.5
TGS/G	*Lr*9,24,26,30,19,25,29/*Lr*1,2a,2c,3a,16,3ka,11,17,20	43.7	56.3	8.5
TRT/G	*Lr*24,19,25,29/*Lr*1,2a,2c,3a,9,16,26,3ka,11,17,30,20	25.0	75.0	12.0
KHT/B	*Lr*1,9,24,19,20,25,29/*Lr*2a,2c,3a,16,26,3ka,11,17,30	43.7	56.3	8.0
TGT/G	*Lr*9,24,26,19,25,29/*Lr*1,2a,2c,3a,16,3ka,11,17,30,20	37.5	62.5	6.5
	Average value	41.7	58.3	11.1
	Akmola region (North)			
TLT/R	*Lr*16,24,26,25/*Lr*1,2a,2c,3,9,3ka,11,17,30,19,20,29	25.0	75.0	8.0
RLP/H	*Lr*2c,16,24,26,11,19,25/*Lr*1,2a,3,9,3ka,17,30,20,29	43.7	56.3	5.0
CJF/B	*Lr*1,2a,2c,9,26,3ka,11,19,20,25,29/*Lr*3,16,24,17,30	68.7	31.3	9.0
THT/J	*Lr*9,24,19,29/*Lr*1,2a,2c,3a,16,26,3ka,11,17,30,20,25	25.0	75.0	12.0
TGT/G	*Lr*9,24,26,19,25,29/*Lr*1,2a,2c,3a,16,3ka,11,17,30,20	37.5	62.5	18.0
TQT/G	*Lr*24,26,19,25,29/*Lr*1,2a,2c,3a,9,16,3ka,11,17,30,20	31.3	68.7	10.0
TGK/G	*Lr*9,24,26,3ka,19,25,29/*Lr*1,2a,2c,3a,16,11,17,30,20	43.7	56.3	8.0
TBT/Q	*Lr*9,16,24,26,25,29/*Lr*1,2a,2c,3a,3ka,11,17,30,19,20	37.5	62.5	8.0
RGT/G	*Lr*2c,9,24,26,19,25,29/*Lr*1,2a,3a,16,3ka,11,17,30,20	43.7	56.3	6.0
TQT/M	*Lr*24,26,20,25/*Lr*1,2a,2c,3a,9,16,3ka,11,17,30,19,29	25.0	75.0	6.0
SBR/H	*Lr*3a,9,16,24,26,17,19,25/*Lr*1,2a,2c,3ka,11,30,20,29	50.0	50.0	5.0
THT/B	*Lr*9,24,19,20,25,29/*Lr*1,2a,2c,3,16,26,3ka,11,17,30	37.5	62.5	5.0
	Average value	39.1	61.0	8.3
	Almaty region (Southeast)			
JCL/G	*Lr*1,3a,9,16,24,11,17,30,19,25,29/*Lr*2a,2c,26,3ka,20	68.7	31.3	28
PBN/C	*Lr*2a,9,16,24,26,11,30,19,20,25/*Lr*1,2c,3,3ka,17,29	62.5	37.5	21
QBQ/G	*Lr*2c,3a,9,16,24,26,17,30,19,25,29/*Lr*1,2a,3ka,11,20	68.7	31.3	22
SBP/C	*Lr*3a,9,16,24,26,11,19,20,25/*Lr*1,2a,2c,3ka,17,30,29	56.3	43.7	29
	Average value	64.1	36.0	25.0

R—resistant response of *Lr* genes to *Ptr* pathotypes; S—susceptible response of *Lr* genes to *Ptr* pathotypes.

**Table 2 plants-10-01484-t002:** The results of assessing spring wheat varieties to the pathotypes of *P. triticina* and identification of leaf rust resistance genes.

Cultivar Name	Origin	Year of Release	Reaction to Infection with Pathotypes *P. triticina*										Leaf Rust Severity %, RT		*Lr* Gene (s) Present Based on Markers
			QBQ/G		SBR/H		KHT/B		SBP/C		THT/B				
			1st Score	2nd Score	1st Score	2nd Score	1st Score	2nd Score	1st Score	2nd Score	1st Score	2nd Score	2019	2020	
Akmola 2	KZ:Shortandy	1998	4+	4+	3−Yes we confirm	4+	3−	4+	4+	4+	3−	4+	20MR	10MR	*Lr* *1*
Almaken	KZ:Almaty-KIZ	2010	4+	4+	3	4+	3−	4	4	4+	3	4+	70S	40MS	*Lr* *1*
Albidum 28	RU:Saratov	1987	3+	4+	3	4+	3−	4	4+	4+	3	4+	40MS	30MS	*Lr10*
Astana	KZ:Shortandy	2004	3	4+	4	4+	3	4	4	4+	4	4+	70S	20MS	*Lr**1*, *Lr6**8*
Astana 2	KZ:Shortandy	2008	4+	4+	4	4+	3	4	3−	4+	3	4+	20MR	10MR	*Lr* *1*
Avangard	KZ:East-VNIISH	2005	3	4+	2−	4	3−	4	2 + 3	4+	0	0	40MS	30MS	none
Bayterek	KZ:Shortandy	2008	3	4+	2−	4+	2−	3	3	4	3−	4	80S	40MS	*Lr* *1* *Lr28*
Zhenis	KZ:Almaty-KIZ	2006	3	4+	3	4	3−	4	4	4+	3−	4	40MS	30MS	*Lr* *1*
Kazakhstanskaya rannespelaya	KZ:Almaty-KIZ-Karabalyk	1991	3−	4+	2−	4+	3−	4+	3	4+	0	0	20MS	10MR	*Lr* *1*
Kazakhstanskaya 10	KZ:Almaty-KIZ	1996	4+	4+	3	4+	3−	4+	4	4+	3−	4+	30MS	20MS	none
Kazakhstanskaya 19	KZ:Almaty-KIZ	1994	3	4+	4	4+	3	4	3	4+	3	4+	20MR	10MR	*Lr**1*, *Lr28*, *Lr**34*
Kazakhstanskaya 25	KZ:Almaty-KIZ	1997	2	2	3+	4+	3	4	4	4+	3−	4+	20MR	15MR	*Lr* *1*
Karabalykskaya 92	KZ:Karabalyk	1997	3−	4+	2−	4	3	4	4+	4+	3	4+	40MS	20MS	none
Karagandunskaya 70	KZ:Karaganda	1992	0	4	3	4	1	3	3−	4+	3−	4	30MS	20MS	*Lr**1*, *Lr6**8*
Lyazzat	KZ:East- VNIISH	2011	1	4	2−	3+	3-	4	3+	4+	3	4+	60S	50S	*Lr* *1*
Nargiz	KZ:East- VNIISH	2011	3−	4	3	4+	2 + 3	4	4−	4+	3	4+	40S	30MS	none
Omskaya 36	RU:Omsk-AC	2007	2	2	3	4	3-	4+	2	2	4	4	70S	50S	none
Pavlodarskaya 93	KZ:Pavlodar	1999	2−	4−	3	4+	3	4+	2−	4+	3	4	90S	30MS	*Lr* *1*
Stepnaya 2	RU:Saratov	2010	4	4	3−	4+	3−	4+	3	4+	4+	4+	40MS	30MS	*Lr* *1*
Ulbinka 25	KZ:East- VNIISH	1989	2 + 3	4−	3+	4+	3	4	3+	4+	3	4	50S	40MS	none
Tselina 50	KZ:Shortandy	2010	4	4	4+	4	3	4+	4+	4+	4	4+	30MS	90S	*Lr* *1*
Tselinnaya 3C	KZ:Shortandy	1996	3	4	4+	4	3+	4+	4+	4+	4+	4+	30MS	100S	*Lr* *1*
Shortandinskaja uluchshennaja 95	KZ:Shortandy	2006	2	3+	1 + 2	4	2 + 3	4+	1 + 2	4	2 + 3	3	40MS	100S	*Lr28*
Erythrospermum 35	KZ:Karabalyk	1991	3	4+	4	4	3	4+	4	4+	4	4+	50S	30MS	*Lr**1*, *Lr6**8*
Erythrospermum 841	RU:Saratov	1942	3	4+	4+	4	3	4+	4+	4+	3+	4	60S	40MS	*Lr* *1*
Stepnaya 62	KZ:Aktyubinsk	SVT 2017	0	2	2	3	0	2+	0	2	0	0	20MR	10MR	*Lr* *1*
Omskaya 37	RU:Omsk-AC	2016	0	2	3	3	0	2+	3	3+	0	0	0	10R	*Lr1*, *Lr10*, *Lr19*, *Lr26*
Tertsiya	RU:Omsk-AC	1995	2	3+	3	4	3−	4	3−	4+	2 + 3	4	0	10R	*Lr1*, *Lr**9*
Chelyaba jubilejnaja	RU:Chelyabinsk	2010	3−	4	3-	4+	3	4+	3−	4	3	4	0	15MR	*Lr1*, *Lr**9*
Saratovskaya 29	Ru:Saratov	1957	3	4+	4	4+	3	4+	4+	4+	4+	4+	60S	50S	*Lr1*, *Lr**10*, *Lr28*
Morocco (susceptible check)	Morocco	-	4	4+	4	4+	4	4+	4+	4+	4+	4+	100S	90S	none

Abbreviations: SVT—cultivar candidate submitted to the state variety testing; RT—reaction type.

## Data Availability

Not applicable.

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
