# Peer review of "Evaluation of Wheat Germplasm for Resistance to Leaf Rust (Puccinia triticina) and Identification of the Sources of Lr Resistance Genes Using Molecular Markers"

_plants, 2021, doi:10.3390/plants10071484_

Round 1

Reviewer 1 Report

I think the manuscript has valuable results about the study of the Puccinia triticina population in  Kazakhstan and for the evaluation of the resistance to the pathogen in a set of 30 spring wheat genotypes from Russia and Kazakhstan. My main concern is about the organization of the manuscript, as materials and methods is not described in the same order as results, which make reading a bit difficult.

Furthermore, a further English editing could improve some parts of the manuscript

Reviewer 2 Report

The work was aimed at investigation on variation in Ptr population from Kazakhstan, where leaf rust epidemics were frequent. Furthermore, a work was aimed at investigation on examination resistance during seedling and adult plant stages and identifycation the sources of Lr 16 resistance genes among the spring wheat collection using molecular markers.

In addition, limited information is available on the Lr genes present in commercial varieties and advanced spring wheat breeding lines from Kazakhstan.

Therefore, I consider the manuscript as novel and interesting and the research question as important, because it allows to identify wheat genotypes highly resistant to leaf rust that can help to improve leaf rust resistance

The comprehensive approach and methods used have led to identify wheat genotypes highly resistant to leaf rust.

The manustript is clear, unambiguous, and well-written.

The article is clearly laid out and are all the key elements are present

The introduction provides comprehensive informations on the background to show the context of the research.

The project is appropriate.  The authors explains how the data was collected.

The article specify procedures and the sampling was appropriate.

The authors explains clearly laid out and in a logical sequence what they discovered in the research. The statistics are correct.  

The claims are supported by the results. The authors indicated how the results relate to expectations and to earlier research.

Lines 61-63 and 98-100: same sentence

Line 256: delete the S1 table title, please

The tibles  are relevant, well labeled and described.

Taking the above into consideration, I recommend this manuscript for publication with a minor revisions.
